# Seasonal and Geographical Impact on the Mycotoxigenicity of *Aspergillus* and *Fusarium* Species Isolated from Smallholder Dairy Cattle Feeds and Feedstuffs in Free State and Limpopo Provinces of South Africa

**DOI:** 10.3390/toxins15020128

**Published:** 2023-02-04

**Authors:** Oluwasola Abayomi Adelusi, Sefater Gbashi, Janet Adeyinka Adebo, Adeola Oluwakemi Aasa, Oluwaseun Mary Oladeji, Glory Kah, Oluwafemi Ayodeji Adebo, Rumbidzai Changwa, Patrick Berka Njobeh

**Affiliations:** 1Department of Biotechnology and Food Technology, Faculty of Science, Doornfontein Campus, University of Johannesburg, P.O. Box 17011, Johannesburg 2028, South Africa; 2Food Evolution Research Laboratory, School of Hospitality and Tourism, College of Business and Economics, University of Johannesburg, P.O. Box 17011, Johannesburg 2028, South Africa; 3Department of Biology and Environmental Science, Faculty of Science, Sefako Makgatho Health Sciences University, P.O. Box 139, Pretoria 0204, South Africa; 4Food Innovation Research Group, Department of Biotechnology and Food Technology, Doornfontein Campus, University of Johannesburg, P.O. Box 17011, Johannesburg 2028, South Africa

**Keywords:** dairy cattle feeds, fungal species, mycotoxigenicity, mycotoxins, LC-MS/MS, South Africa

## Abstract

This study evaluated the impact of seasonal and geographical variations on the toxigenicity of *Aspergillus* and *Fusarium* strains previously isolated from smallholder dairy cattle feeds and feedstuffs sampled during summer and winter in the Free State and Limpopo provinces of South Africa (SA). In total, 112 potential toxigenic fungal species were obtained and determined for their capability to produce mycotoxins on solid Czapek Yeast Extract Agar (CYA); followed by liquid chromatography-mass spectrometry (LC-MS/MS) analysis. Our result revealed that 41.96% of the fungal species produced their respective mycotoxins, including aflatoxin B_1_ (AFB_1_), aflatoxin B_2_ (AFB_2_), and zearalenone (ZEN), with higher levels of AFB_1_ (0.22 to 1045.80 µg/kg) and AFB_2_ (0.11 to 3.44 µg/kg) produced by fungal species isolated from summer samples than those in winter [(0.69 to 14.44 µg/kg) and (0.21 to 2.26 µg/kg), respectively]. The same pattern was also observed for AFB_1_ and AFB_2_ in Limpopo (0.43 to 1045.80 µg/kg and 0.13 to 3.44 µg/kg) and Free State (0.22 to 576.14 µg/kg and 0.11 to 2.82 µg/kg), respectively. More so, ZEN concentrations in summer (7.75 to 97.18 µg/kg) were higher than in winter (5.20 to 15.90 µg/kg). A similar observation was also noted for ZEN in Limpopo (7.80 to 97.18 µg/kg) and Free State (5.20 to 15.90 µg/kg). These findings were confirmed via Welch and Brown-Forsythe tests with significantly (*p* ≤ 0.05) higher mycotoxin levels produced by fungal strains obtained in samples during summer than those in winter. In contrast, the concentrations of mycotoxins produced by the fungal species from both provinces were not significantly (*p* > 0.05) different.

## 1. Introduction

Feed refers to any unprocessed or processed substance consumed by livestock to meet their nutritional demands, whereas feedstuff is the raw material from vegetable or animal sources required to formulate compound feeds for animal consumption. The proportion of essential components in livestock feeds differs based on the species, physiological status, age, and sex of the animal [1]. Dairy cattle are given the required diets to supply them with the essential nutrients mainly for milk production and dietary requirements vary significantly based on the gestation and lactation stages [2]. Another important aspect of dairy feed is its safety. In general, feed safety is a prerequisite for the safety of food of animal origin, particularly dairy products. Unfortunately, a majority of dairy cattle farmers in sub-Saharan African (SSA) nations, including South Africa (SA), lack the basic knowledge of feed safety, good farm management practices, and animal nutrition [3,4]. Dairy farmers in this region primarily focus on increasing productivity, with no concern for milk and milk product safety, often compromised when dairy cows are fed contaminated feeds. Among dairy cattle feed contaminants, mycotoxins are a severe threat to the dairy industry in SA [5,6].

Mycotoxins are toxic substances formed by toxigenic fungi which infect agricultural products on the farm or during storage under certain conditions, including improper storage, high relative humidity, high temperature, and insect and pest infestation [7,8]. These contaminants when found in dairy feeds and feed ingredients, especially at high contamination levels, might pose a serious threat to milking cows. Firstly, they may have a detrimental impact on dairy cattle health, such as decreased feeding efficiency, reduced milk yield, immunosuppression, weight loss, laminitis, low fertility, and abortion [2,9,10]. Secondly, they may compromise the food supply chain as a result of their transfer from feed to milk and milk products [11,12,13] owing to their tolerance to high temperatures [14], making milk production operations inadequate and ineffective for their complete elimination during milk production and processing [15,16,17]. As a result, humans are exposed to these toxins through the consumption of contaminated meat, milk, and milk products. 

Human exposure to mycotoxins causes various health effects as they are known to be nephrotoxic, hepatotoxic, teratogenic, carcinogenic, and immunosuppressive [9,18,19,20]. It may result in death in dire cases, such as prolonged chronic toxicity or high acute intoxication [18]. In addition to their negative impacts on animal and human health, mycotoxins also cause significant economic losses for many nations, particularly those in SSA, because of the costs directed toward food safety [21].

The most prevalent toxigenic fungi in agricultural commodities are species belonging to the *Aspergillus*, *Alternaria*, *Fusarium*, and *Penicillium* genera [22,23]. Among them, *Fusarium*, *Aspergillus*, and *Penicillium* are considered the greatest producers of mycotoxins in cow diets worldwide [24,25,26,27], including SA [28,29,30]. Also, fungal toxins like aflatoxins (AFB_1_, AFB_2_, AFG_1_, and AFG_2_) formed by fungal species in the genera *Aspergillus*; ochratoxins (OTA and OTB) produced by *Penicillium* and *Aspergillus* spp.; and deoxynivalenol (DON), trichothecenes (T-2 and HT-2 toxins), zearalenone (ZEN), and fumonisins (FB_1_ and FB_2_) formed by *Fusarium* spp. are significant mycotoxins found regularly in dairy feeds globally [5,31,32], and SA [6,33].

There is growing evidence to suggest that seasonal and geographical differences influence mycotoxin formation in food and feed [34,35]. Unfortunately, there are limited studies in SA to demonstrate this, particularly in dairy feeds and feed components. For the most part, past studies conducted in the country only revealed the prevalence of mycotoxins in dairy feeds and feedstuffs [33,36,37], while others only recognized fungi as the causative agents for the contamination [28,38]. Therefore, this study aims to screen and quantify the extent of mycotoxin production by the toxigenic fungal species invading smallholder dairy feeds in SA, based on seasonal (summer and winter) and geographical (Limpopo and Free State) variations. The findings from this study could be a significant reference for mitigating and preventing mycotoxin contamination in dairy feeds and, subsequently, in human diets derived from dairy cattle in SA.

## 2. Results

### 2.1. Method Validation

Mycotoxins synthesized by *Fusarium* and *Aspergillus* strains in this research were identified and quantified by means of liquid chromatography-mass spectrometry (LC-MS/MS). The mycotoxins tested for in this study include AFB_1_, AFB_2_, AFG_1_, AFG_2_, OTA, ZEN, and DON. Standard validation parameters for each matrix were achieved based on recovery, retention time, linearity, the limit of detection (LOD), and the limit of quantification (LOQ). As seen in Table 1, the analytical technique revealed linearity with R^2^ for all mycotoxin levels varying from 0.9966 to 0.9995. Furthermore, the LODs and LOQs of various mycotoxins varied from 0.01 to 4.42 and 0.04 to 13.40 µg/kg, respectively, whereas the mean recoveries varied from 71.4 to 101.9%, falling within the European Commission (EU) acceptable performance criteria [39]. Figure 1 represents the multiple reaction monitoring (MRM) chromatograms of AFs (AFB_1_ and AFB_2_) produced by *A. flavus* and ZEN formed by *F. oxysporum* in contrast to corresponding mycotoxin standards.

### 2.2. Toxigenicity of Aspergillus and Fusarium Species

Data on the mycotoxigenicity of the fungal isolates are summarized in Table 2 and Table 3 as well as in Appendix A. About 41.96% of the total 112 fungal species were positive, producing their respective mycotoxins (AFB_1_, AFB_2_, and ZEN). Furthermore, 51.47 and 27.27% of the fungal isolates recovered during summer and winter, as well as 37.5 and 47.92% of those recovered from Free State and Limpopo feeds tested positive for the analyzed mycotoxins. More so, our results revealed that *A. flavus* was the sole AF producer among the 83 *Aspergillus* species isolated from the feed samples, producing only AF B-types (AFB_1_ and AFB_2_). Of the 29 *A. flavus* strains examined in this work, 82.76 and 37.93% were positive for AFB_1_ and AFB_2_, respectively. However, 90 and 45% of summer *A. flavus* strains produced AFB_1_ and AFB_2_. Similarly, 66.67 and 22.22% of winter *A. flavus* strains produced the same mycotoxins. It is worth noting that 75 and 31.25% of *A. flavus* from Free State samples, as well as 92.31 and 46.15% of *A. flavus* from Limpopo, were positive for AFB_1_ and AFB_2_, respectively. Interestingly, none of the strains of *A. ochracheus* and *A. niger*, the well-known producers of OTA, produced mycotoxin. Also, no strain of *A. fumigatus* in this study produced any toxin.

In addition, ZEN, the sole *Fusarium* mycotoxin found in this work, was primarily formed by strains of *F. oxysporum* and *F. equiseti*, with prevalence rates of 63.64 and 71.43%, respectively. It must be emphasized that 3/3 (100%) and 2/4 (50%) of the summer and winter *F. equiseti* produced ZEN, whereas 2/2 (100%) and 3/5 (60%) of the same isolate from Free State and Limpopo feed samples were toxigenic with respect to this toxin. Among the *F. oxysporum* strains examined, 5/8 (62.5%) and 2/3 (66.67%) from summer and winter samples, and 5/8 (62.5%) and 2/3 (66.67%) of those from Free State and Limpopo tested positive for ZEN. Meanwhile, all *Fusarium* spp. failed to produce DON.

As shown in Table 2 and Table 3 and Figure 2, the concentrations of AFB_1_ produced by *A. flavus* in summer (range: 0.22–1045.80; mean: 127.01 µg/kg) were higher than in winter (range: 0.69–14.44; mean: 3.85 µg/kg). AFB_2_ showed the same trend in summer (range: 0.11–3.44; mean: 1.40 µg/kg) and winter (range: 0.21–0.26; mean: 0.24 µg/kg). Additionally, ZEN concentrations formed by both *F. oxysporum* and *F. equiseti* strains were higher in summer (range: 7.75–97.18; mean: 22.55 μg/kg) than in winter (range: 5.20–15.90; mean: 9.46 μg/kg). Furthermore, AFB_1_ levels produced by *A. flavus* from Free State (range: 0.22–576.14; mean: 70.47 µg/kg) were lower than those recorded in *A. flavus* from Limpopo (range: 0.43–1045.80; mean: 122.13 µg/kg). A similar pattern was also seen in AFB_2_ levels formed by *A. flavus* from Free State (range: 0.11–2.82; mean: 1.14) and Limpopo (range: 0.13–3.44; mean: 1.22 μg/kg). ZEN concentrations produced by *F. oxysporum* and *F. equiseti* strains from Free State samples (range: 5. 20–15.90; mean: 9.83 μg/kg) were lower than the concentrations of ZEN produced by both fungal strains from Limpopo samples (range: 7.80–97.18; mean: 29.88 μg/kg). 

### 2.3. Seasonal and Geographical Impacts on Mycotoxigenicity of Aspergillus and Fusarium Species

Table 4 and Table 5 showed the Welch and Brown-Forsythe robust test for the equality of means for AFB_1_ and AFB_2_ biosynthesis by *A. flavus*, as well as ZEN formation by *F. oxysporum* and *F. equiseti* in SA smallholder dairy cattle feeds and feed ingredients with respect to seasons (summer and winter) and locations (Free State and Limpopo). The results showed that season had a significant effect (*p* ≤ 0.05) on all the mycotoxins (AFB_1_, AFB_2,_ and ZEN) formed by the toxigenic fungal strains, while the potential of the fungal isolates to produce mycotoxins in both locations was not statistically different (*p* > 0.05).

## 3. Discussion

Animal feed contamination by microorganisms, especially toxigenic fungal species is a worldwide concern owing to the mycotoxins they produce in agricultural products, which can be harmful to human and animal health [40] with a serious impact on the economy of any nation [21]. Mycotoxin contamination greatly contributes to agricultural product loss in SSA [41]. The prevalence of mycotoxin-producing fungi, particularly, *Fusarium* and *Aspergillus*, have previously been documented in feeds destined for smallholder dairy cows in both Free State and Limpopo provinces of SA [30], implying the presence of the mycotoxins recovered and reported herein.

In the present study, a total of 112 *Fusarium* and *Aspergillus* strains obtained from smallholder dairy cow diets across different seasons (summer and winter) from Free State and Limpopo provinces of SA were investigated for their mycotoxin-producing potentials. The study also investigated the impact of variation in seasonal and geographical locations on the mycotoxin production capacity of the fungal strains. *A. flavus* strains analyzed in this present work produced only B-type AFs (AFB_1_ and AFB_2_) but none of the G-types (AFG_1_ and AFG_2_). These findings are in line with those reported by Chilaka et al. [38], Lasrem et al. [42], and Yogendrarajah et al. [43], based on the formation of only AFB_1_ and AFB_2_ by strains of *A. flavus* recovered from various agricultural commodities from SA, Sri Lanka, and Tunisia, respectively. It must be emphasized that the levels of AFs produced by the *A. flavus* strains in this study ranged from 0.22 to 1045.80 µg/kg, wherein 9/24 (38%) of the fungal species produced AF levels above the SA maximum tolerable levels (10 µg/kg) for dairy cattle feeds [5]. We also found that all the *A. fumigatus*, *A. niger,* and A. *ochraceus* strains reported in this work failed to produce any AF analog.

*A. niger* has been documented as one of the major OTA producers of agricultural products [44,45]. However, OTA production by *Aspergillus* spp. occurred under different environmental conditions, including temperature, water activity (a_w_), and pH [46], as well as the carbon source [47]. Wei et al. [47] confirmed that arabinose was the optimum carbon source for OTA production in *A. niger* CBS 513.88, followed by glucose and sucrose. Nevertheless, none of the 25 strains of *A. niger* examined in this study produced OTA. This disparity might be because the *A. niger* strains lack the genetic potential to synthesize OTA. Susca et al. [48] investigated several OTA-producing and non-producing *A. niger* strains recovered from various agricultural commodities such as maize, pistachio, walnuts, and cashew nuts from several countries (Turkey, Italy, USA, Brazil, Argentina, China, Greece, Indian, Iran, and Spain), and discovered that the non-OTA-producing strains lacked OTA biosynthetic cluster genes (*ota1*, *ota2*, *ota3*, and *ota5*), with further genome sequence analysis data revealing OTA gene deletion in the non-OTA-producing *A. niger* isolates. More so, the culture substrate (CYA) utilized in this study could be linked to the *A. niger* strains’ non-ochratoxigenic nature, as Esteban et al. [49] observed that mean OTA levels produced by *A. carbonarius* and *A. niger* are substrate dependent. Our findings regarding OTA production by *A. niger* are in agreement with that observed by Munitz et al. [50], who found that none of the 19 *A. niger* strains obtained from Argentinian blueberry produced OTA, similar to the study conducted in Africa by Njobeh et al. [51] where over 17 strains of *A. niger* isolated from Cameroonian food commodities were non-OTA producers. Our results were also in line with the study performed by Sultan and Magan [52], wherein none of the *A. niger* strains recovered from peanuts sourced from different geographical regions of Egypt tested positive for mycotoxin. Chilaka et al. [38] also conducted similar research in SA and found that the 10 strains of *A. niger* isolated from maize, an essential dairy cattle feed ingredient in the country, could not produce OTA. However, this result is unsurprising as earlier reports on OTA contamination of dairy feed in SA revealed that the mycotoxin was not detected in the feed samples [36,53]. *A. ochraceus*, another notable OTA-producing fungus [38,54], isolated from the feed samples, was not ochratoxigenic, and as conjectured, neither the experimental conditions, including the culture medium and laboratory conditions (temperature, a_w_, and pH) were ideal for OTA biosynthesis by the fungal species or the isolates lacked the required genes for OTA production. According to Chilaka et al. [38] and Wang et al. [54], the fungal species metabolic profiling relies on the laboratory conditions and the growing medium, while O’Callaghan et al. [55] confirmed that the expression of the biosynthetic genes (polyketide synthase) involved in OTA biosynthesis by *A. ochraceus* is highly regulated by nutritional cues and pH. Moreover, our findings on OTA production by both *A. ochraceus* and *A. niger* contradict those of Rosa et al. [56], who reported that the two fungi isolated from dairy cow feed in the Brazilian state of Rio de Janeiro produced OTA. Adekoya et al. [57] also demonstrated that 87% of *A. niger* isolates from Nigerian and SA fermented food products tested positive for OTA.

Among the *Fusarium* spp. tested in this work, only cultures of *F. oxysporum* and *F. equiseti* were positive for one of the tested mycotoxins (ZEN). This result concurs with the research conducted on mosquitoes in SA by Phoku *et al*. [58]. Barros et al. [59] also confirmed ZEN formation by *F. equiseti* in Argentinean soybean. However, ZEN concentrations reported in this present work were low, ranging from 5.20 to 97.18µg/kg, a range far below the SA acceptable limit (500 µg/kg) in dairy cattle feeds [5]. Nevertheless, earlier reports have indicated that ZEN is more prevalent in Western Europe, North America, and Eastern Europe than in African countries [60]. However, our results disagree with the report of Chilaka et al. [38] wherein none of the *F. oxysporum* isolated from SA maize tested positive for ZEN. Another important *Fusarium* toxin, DON, was detected in this study as none of the tested fungal isolates could produce the toxin. According to Neme and Mohammed [61], DON is primarily formed by *Fusarium spp.,* in particular, *F. culmorum* and *F. graminearum*. Also, fumonisin B_1_ (FB_1_) and fumonisin B_2_ (FB_2_) could not be confirmed in this study due to the lack of reference materials.

Data on the impact of seasonal and geographical variations on the toxigenic potentials of fungal species invading dairy cattle diets and feed raw materials is scarce worldwide, including in SA. In terms of seasonal variation, significant (*p* ≤ 0.05) effects were noted on the biosynthesis of AFB_1_ and AFB_2_ by *A. flavus* and ZEN by *F. oxysporum* and *F. equiseti*, in which higher levels of the mycotoxins were formed by the summer fungal species in comparison to the winter fungal isolates. These findings are in excellent agreement with what Mohale et al. [62] reported on mycotoxin production by toxigenic fungal species associated with stored maize (essential dairy feed ingredients) from various parts of Lesotho, with findings revealing mean concentrations of AFB_1_ (73, 338 µg/kg) and AFB_2_ (3, 117 µg/kg) produced by *A. flavus* in 2009/10 cropping season being significantly higher than AFB_1_ (821 µg/kg) and AFB_2_ (0 µg/kg) produced during 2010/11 cropping season. Interestingly, our investigation, on the contrary, revealed that location had no significant (*p* > 0.05) effect on the toxigenicity of the fungal isolates. Our results on the effect of location on the fungal mycotoxin profiling agree with those of Yazid et al. [44], who found that agro-ecological location has no influence on AFB_1_ formation by *A. flavus* invading grain corn farms (Kampong Dadong and Rhu Tapai) in Terengganu, Malaysia. The authors emphasized that the mycotoxigenicity of the fungal isolates is more dependent on fungal strain specificity than geographical locations. 

The ability of the isolated fungal species to produce different mycotoxins is a concern for the welfare of dairy cattle in the areas, as earlier mycotoxin analysis of similar feeds from the two provinces revealed the presence of all the attendant mycotoxins (AFB_1_, AFB_2_, and ZEN) reported in this study, in addition to OTA, DON, AFG_1,_ and AGF_2_, which were not produced by the toxigenic microbiota [6]. However, the study confirmed a low incidence of OTA, AFG_1,_ and AFG_2_ [6]. As motioned previously, the failure of the tested fungal isolates to produce these toxins may be linked to the culture medium [38], strain specificity [44], or the absence of mycotoxin cluster genes in the fungal species [55].

The high level of mycotoxins formed by both *Fusarium* and *Aspergillus* strains in summer feeds in this work implies that fungal species produce high levels of mycotoxin in summer than in winter. This is likely attributed to climatic differences since toxigenic fungal colonization of crops and their ability to produce mycotoxin is dependent on climatic conditions [7,63]. Temperature, rainfall, and relative humidity are the primary climatic parameters promoting fungal growth and mycotoxin production in food and feed [63]. Even though fungal proliferation and mycotoxin biosynthesis are linked, the ideal relative humidity and temperature for mycotoxin biosynthesis differ based on the fungus and its associated mycotoxins [64]. Although *A. flavus* spp. can produce AFs under a wide range of temperature conditions, an ideal range for their maximum production is between 25 to 35 °C [65]. At high temperatures, more AFB is synthesized than AFG, but the biosynthesis of both toxins is surmised to be similar at low temperatures [66]. Limpopo and Free State areas of SA are usually hot and humid during summer but cold and mild during winter. Free State summer temperatures range from 15 to 31 °C, and winter temperatures range from 1 to 18 °C, whereas summer temperatures in Limpopo range from 18 to 38 °C and a winter temperature range from 12 to 24 °C. The higher temperature conditions observed during summer than in winter in the two provinces must have been the reason for the increased AF production by *A. flavus* strains recovered from the summer samples.

It has been demonstrated that relative humidity levels ranging from 88 to 95% promote fungal proliferation and subsequent toxin production in agricultural products [67]. According to Muga et al. [68], 90% relative humidity greatly enhances AF formation in SA-stored maize kernels. These conditions are identical to the ambient climatic parameters often encountered during summer in both Limpopo and Free State areas of SA and hence account for the elevated AFs and ZEN concentrations produced by summer fungal isolates that did not differ significantly (*p* > 0.05) between the two provinces. The higher mycotoxin concentrations produced by the toxigenic fungal species during summer may also be attributed to post-harvest conditions, such as improper feed management and storage practices [69].

## 4. Conclusions

In conclusion, the present study revealed variations in mycotoxin levels produced by toxigenic fungal isolates previously recovered from contaminated smallholder dairy cows’ feeds and feed raw materials sourced during two different seasons (summer and winter) and geographical regions (Limpopo and Free State) of SA. The high formation of AFB_1_ and AFB_2_ by *A. flavus* strains, as well as ZEN by *F. oxysporum* and *F. equiseti* in this study, demonstrate that similar feeds are likely contaminated by attendant mycotoxins. It is equally important to necessitate feed safety awareness-raising programs specifically on fungal and mycotoxin contamination among various smallholder dairy cattle farmers in SA. More so, dairy cattle farmers in both regions under study must be informed of eco-physiological conditions that enhance fungal proliferation and mycotoxin formation in dairy cow feeds. Also, farmers and consumers must be educated on the risk of carry-over of these toxins from feeds to meat, milk, and other dairy products, as well as the associated health implications in animals and humans.

It thus, follows that data generated in this study should lay the foundation for monitoring the mycotoxins in food and feed that can necessitate particular attention, especially for feed during the summer. However, the present results might not accurately reflect the general scenario of the mycoflora toxigenicity in the diary feeds because of the small sample size and a small proportion of evaluated toxigenic fungal isolates. To fully comprehend the impact of seasonal and geographic variations on the toxigenic potential of mycobiota associated with SA smallholder dairy cattle feeds, a broader survey of mycotoxin-producing fungi with a large sample size that includes year-to-year climatic variations and diverse agro-ecological zones are required.

## 5. Materials and Methods

### 5.1. Chemicals and Reagents

Formic acid, acetonitrile, LC-MS grade methanol, and dichloromethane used in this study were procured from Merck (Darmstadt, HE., Germany). 

### 5.2. Mycotoxin Standards

Standards used for Aspergillus mycotoxins quantification, including OTA, AFB_1_, AFB_2_, AFG_1_, and AFG_2_, and those used for quantifying Fusarium toxins (DON and ZEN), were supplied by Sigma-Aldrich (Steinheim, NRW., Germany). 

### 5.3. Fungal Strains and Inoculation

Pure fungal strains (*n* = 112) used in this study, including *Aspergillus* (83) and *Fusarium* (29), were recovered from smallholder dairy cow feeds collected from various dairy farms in Limpopo and Free State provinces in SA between summer 2018 and winter 2019. Information on fungal isolation and identification can be found in Adelusi et al. [30]. The fungal isolates were tested for their potential to produce mycotoxins, including AFs (AFB_1_, AFG_1_, AFB_2,_ and AFG_2_), DON, OTA, and ZEN by inoculating them on solid CYA agar (Appendix B contains the procedures for agar preparation) treated with streptomycin and chloramphenicol to prevent bacterial growth. Following inoculation, the CYA plates were cultured in the dark for 21 days at 25 °C according to the method of Adekoya et al. [57].

### 5.4. Multi-Mycotoxin Extraction

Extraction of AFs from *Aspergillus* isolates was performed following the agar plus method described by Adekoya et al. [57] with slight modifications. One gram of pure colonies (together with the medium) was gently removed from the CYA plates and transferred to an amber vial containing 4 mL of dichloromethane. The content was vortexed for about 2 min, left for 60 min, and then filtered into a 1.5 mL amber vail using a Milex syringe filter (0.22 µm). ZEN was extracted from *Fusarium* isolates using the method reported by Adekoya et al. [57]. Briefly, 10 g of individual isolate with the medium was transferred into a 250 mL conical flask, and 50 mL of acetonitrile: water (60/40, *v/v*) as extraction solvent was added. The content was agitated for 60 min before being filtered via a Whatman #4 filter paper (Merck, Johannesburg, South Africa), and pH was adjusted to 6.2 ± 0.3 by adding 1 M H_2_SO_4_. After that, the filtrate was extracted thrice using 25 mL of dichloromethane in a 250 mL separating funnel. Following this, acetonitrile (25 mL) was added to the dichloromethane-extracted content and passed over a thin layer of sodium sulfate anhydrous to remove excess moisture. Both *Aspergillus* and *Fusarium* extracts were dried using a stream of nitrogen (N_2_) gas and the content was stored at 4 °C prior to analysis.

### 5.5. LC-MS/MS Apparatus and Condition 

The dried extracts were reconstituted in 1500 µL of LCMS-grade methanol (MeOH). Each aliquot (750 µL) was transferred into an amber vial, diluted with an equal volume of dilution solvent that consisted of MeOH and acetonitrile (ACN) (50:50, *v*/*v*), and 5 µL was injected into an LC-MS/MS [70]. The detection and quantification of AFs and ZEN produced by the fungal isolates were achieved on a Shimadzu LC-MS/MS 8040 instrument (Shimadzu Corporation, Tokyo, Japan), which comprised an LC-30AD Nexera chromatograph attached to a CTO-20 AC Prominence Column Oven and SIL-30 AC Nexera autosampler. The analytes were separated chromatographically by passing extracts through a RaptorTM ARC-18 (2.7 UM, 2.1 × 100 mm) column (Restek Corporation, Bellefonte, PA, USA). Solvents A (0.1% formic acid in deionized water) and B [0.1% formic acid in ACN and MeOH (50:50, *v*/*v*)] were used as mobile phases.

Mobile phases A and B were delivered at 0.2 mL/min constant flow rate and 400 bar maximum pressure. The gradient elution operation started with 0.1 min at 10% solvent B, ramped to 95% B in 8.4 min, and maintained for 3 min. The column was re-equilibrated for 1 min using 10% solvent B before the start of the next run, which lasted for 4.5 min, bringing the entire run time to 17 min. After chromatographic separation, all toxins were identified and quantified utilizing a Shimadzu 8040 triple-quadrupole MS 8040 (Shimadzu Corporation, Kyoto, Japan) operated in a positive ionization mode using an electron spray ionization (ESI+) source. The desolvation line (DL) temperature was set at 250 °C, the interface nebulizing gas flow rate was 3 L/min, the heat block temperature was set at 400 °C, and the drying gas flow rate was set at 15 L/min. Finally, data were acquired using the MRM method operated using optimized MS conditions for the analytes (Table 6). The obtained data were processed and visualized by means of Shimadzu LabSolutions software.

### 5.6. Method Validation

Several European Commission criteria [39] were evaluated to determine the method performance in this study. A multi-mycotoxin analytical technique used was validated using blank CYA media. The evaluated parameters included matrix effect, linearity, recovery, LOD, and LOQ. Both matrix-matched and neat standard calibration curves were constructed to investigate matrix effects on the analyzed samples. The levels of mycotoxins produced by the toxigenic fungal isolates were quantified using matrix-matched calibration (MMC) curves. Furthermore, the linearity was determined by spiking a blank CYA medium at seven (7) concentrations using the matrix-matched calibration curve. The calibration curves were generated by plotting the analyte peak areas (y) against the corresponding concentrations (x). The calibration curve was fitted using linear regression.

Retention times (RT) and coefficient of determination (R^2^) were obtained for individual mycotoxins. Both LOD and LOQ for each mycotoxin were also estimated using MMC. The LODs were calculated as 3.3 times the ratio of the residual standard deviation to the slope (Equation (1)), and LOQs were calculated as the concentration equivalent to 10 times the ratio of the residual standard deviation to the slope (Equation (2)) according to Shrivastava and Gupta [71]. All detected mycotoxins (AFB_1_, AFB_2,_ and ZEN) were quantified by comparing their peak area on the calibration curve to the corresponding mycotoxin standards. In addition, the apparent recovery for individual fungal toxins was determined by spiking blank (CYA) media at 50 µg/kg (low) and 100 µg/kg (high) and comparing the observed concentrations and spiked concentrations following extraction as described in Equation 3 by Tebele et al. [72].
(1)$$\mathrm{LOD}=\frac{\mathrm{residual}\;\mathrm{standard}\;\mathrm{deviation}\;\mathrm{of}\;\mathrm{the}\;\mathrm{regression}\;\mathrm{line}}{\mathrm{Slope}}\times 3.3$$
(2)$$\mathrm{LOQ}=\frac{\mathrm{residual}\;\mathrm{standard}\;\mathrm{deviation}\;\mathrm{of}\;\mathrm{the}\;\mathrm{regression}\;\mathrm{line}}{\mathrm{Slope}}\times 10$$
(3)$$\mathrm{Recovery}=\frac{\mathrm{measured}\;\mathrm{concentration}}{\mathrm{spiked}\;\mathrm{concentration}}\times 100$$

### 5.7. Data Analysis

The IBM Statistical Package for SPSS version 27 (IBM Corp., Armonk, NY, USA, 2021) was employed to analyze the data. A one-way analysis of variance (ANOVA) was used to evaluate the possible differences among the mycotoxin levels produced based on season and location. The Levene test was used to assess the equality of variances among the mycotoxin concentrations produced across the two seasons and locations. Lastly, Welch and Brown-robust Forsythe’s test for equality of means was performed wherein the equality of variances could not be established. Differences were considered significant if *p* was ≤0.05.

## Figures and Tables

**Figure 1 toxins-15-00128-f001:**
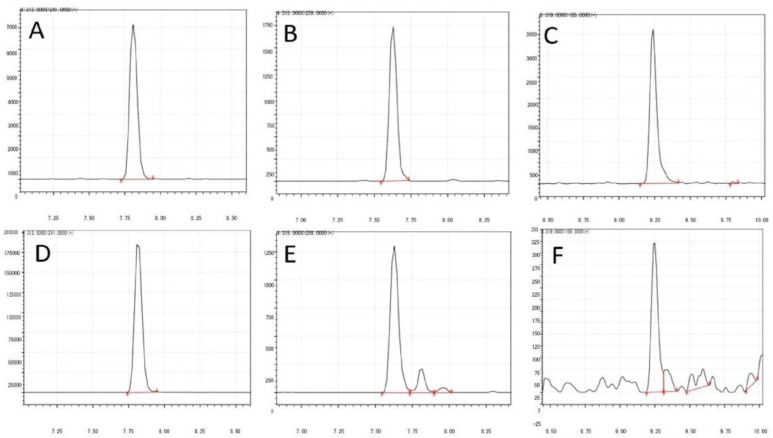
Multiple reaction monitoring (MRM) chromatograms; (**A**): AFB_1_ standard; (**B**): AFB_2_ standard; (**C**): ZEN standard; (**D**, **E**): AFB_1_ and AFB_2_ formed by *A. flavus*; (**F**): ZEN produced by *F. equiseti* isolated from smallholder dairy cattle diets in Limpopo and Free State provinces of SA.

**Figure 2 toxins-15-00128-f002:**
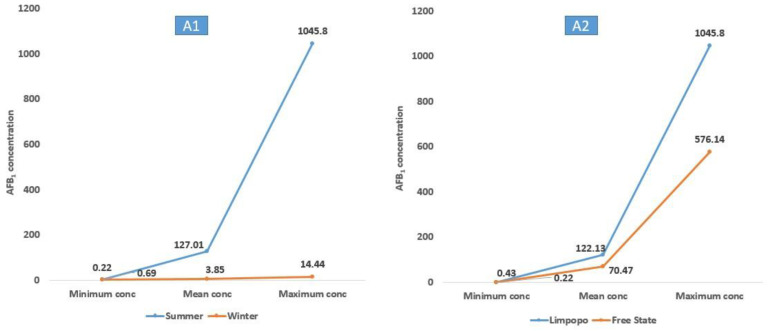
Minimum, mean, and maximum mycotoxin levels (µg/kg) produced by fungal isolates recovered from SA dairy cow diets during the summer and winter seasons; A1 and A2: AFB_1_ produced by *A. flavus* during summer and winter in Free State and Limpopo; B1 and B2: AFB_2_ produced by *A. flavus* during summer and winter and Free State and Limpopo; C1 and C2: ZEN produced by *F. oxysporum* and *F. equiseti* during summer and winter in Free State and Limpopo.

**Table 1 toxins-15-00128-t001:** The matrix-matched calibration curve parameters, recovery, LOD, and LOQ values for CYA.

Mycotoxins	Calibration Points	Ret. Time (min)	R^2^	Slope	% Recovery	LOD (µg/kg)	LOQ (µg/kg)
OTA	25, 50, 250, 500	9.30	0.9987	1864.73	98.3	0.01	0.04
AFB1	0.5, 1, 50, 250	7.84	0.9986	657.99	80.9	0.04	0.14
AFB2	0.5, 1, 100, 250	7.64	0.9988	965.93	101.9	0.02	0.07
AFG1	1, 10, 50, 500	7.45	0.9995	748.77	90.3	0.06	0.19
AFG2	1, 10, 25, 250	7.25	0.9994	420.65	93.3	0.05	0.17
ZEN	0.5, 25, 50, 250	7.75	0.9966	11.09	92.9	0.74	2.24
DON	1, 10, 100, 250	4.90	0.9970	5.45	71.4	4.42	13.40

ZEN: zearalenone; OTA: ochratoxin A; AFB_1_: aflatoxin B_1_; AFB_2_: aflatoxin B_2_; AFG_1_: aflatoxin G_1_; AFG_2_: aflatoxin G_2_; DON: deoxynivalenol; R^2^: Coefficient of determination; Ret: Retention; LOD: limit of detection; LOQ: limit of quantification; CYA: Czapek yeast extract agar.

**Table 2 toxins-15-00128-t002:** The concentrations of mycotoxins (range and mean: µg/kg) produced by *Aspergillus* and *Fusarium* spp. recovered from smallholder dairy cattle feeds sourced during the summer and winter seasons from Limpopo and Free State, SA.

		Summer				Winter			
Isolated Species	Accession No	No of Iso. spp.	No of Tox. spp.	Toxin Produced (Range: µg/kg)	Mean (µg/kg)	No of Iso. spp.	No of Tox. spp.	Toxin Produced (Range: µg/kg)	Mean (µg/kg)
*Aspergillus*									
*A. Flavus*	ON988996	20	18 9	AFB_1_ (0.22–1045.80) AFB_2_ (0.11–3.44)	127.01 1.40	9	6 2	AFB_1_ (0.69–14.44) AFB_2_ (0.21–0.26)	3.85 0.24
*A. fumigatus*	ON988172	15	-	-	-	15	-	-	-
*A. niger*	ON988183	14	-	-	-	8	-	-	-
*A. ochraceus*	ON988182	1	-	-	-	1	-	-	-
*Fusarium.*									-
*F. chlamydosporum*	ON993228	7	-	-	-	4	-	-	
*F. equiseti*	ON991743	3	3	ZEN (8.69–97.18)	41.64	4	2	ZEN (7.64–9.08)	8.36
*F. oxysporum*	ON991521	8	5	ZEN (7.75–16.29)	11.09	3	2	ZEN (5.20–15.90)	10.55
Total		68	35			44	12		

No.: number; tox: toxigenic; iso: isolated; spp: species.

**Table 3 toxins-15-00128-t003:** The concentration of mycotoxins (range and mean: µg/kg) produced by *Aspergillus* and *Fusarium* spp. recovered from smallholder dairy cattle feeds in Free State and Limpopo, SA.

		Free State				Limpopo			
Isolated Species	Accession No	No of Iso. spp.	No of Tox. spp.	Toxin Produced (Range: µg/kg)	Mean (µg/kg)	No of Iso. Spp.	No of Tox. spp.	Toxin Produced (Range: µg/kg)	Mean (µg/kg)
*Aspergillus*									
*A. Flavus*	ON988996	16	12 5	AFB_1_ (0.22–576.14) AFB_2_ (0.11–2.82)	70.47 1.14	13	12 6	AFB_1_ (0.43–1045.80) AFB_2_ (0.13–3.44)	122.13 1.22
*A. fumigatus*	ON988172	16	-	-	-	14	-	-	-
*A. niger*	ON988183	12	-	-	-	10	-	-	-
*A. ochraceus*	ON988182	2	-	-	-	-	-	-	-
*Fusarium*									-
*F. chlamydosporum*	ON993228	8	-	-	-	3	-	-	
*F. equiseti*	ON991743	2	2	ZEN (7.64–8.69)	8.12	5	3	ZEN (9.08–97.18)	41.77
*F. oxysporum*	ON991521	8	5	ZEN (5.20–15.90)	10.49	3	2	ZEN (7.80–16.29)	12.05
Total		64	24			48	23		

No.: number; tox: toxigenic; iso: isolated; spp: species.

**Table 4 toxins-15-00128-t004:** Welch and Brown-Forsythe test showing the effects of seasons (summer and winter) on the formation of AFB_1_ and AFB_2_ by *A. flavus* and ZEN by *F. oxysporum* and *F. equiseti* isolated from SA smallholder dairy cattle feeds.

Mycotoxins	Statistics ^a^	df1	df2	*p*-Value
AFB_1_	11.363	1	53.109	0.001
AFB_2_	23.893	1	26.265	0.001
ZEN	5.813	1	24.898	0.024

a: asymptotically F distributed.

**Table 5 toxins-15-00128-t005:** Welch and Brown-Forsythe test showing the effects of locations (Free State and Limpopo) on the formation of AFB_1_ and AFB_2_ by *A. flavus* and ZEN by *F. oxysporum* and *F. equiseti* isolated from SA smallholder dairy cattle feeds.

Mycotoxins	Statistics ^a^	df1	df2	*p*-Value
AFB_1_	0.840	1	54.639	0.363
AFB_2_	0.035	1	29.230	0.852
ZEN	4.148	1	14.359	0.061

a: asymptotically F distributed.

**Table 6 toxins-15-00128-t006:** Mass spectrometric parameters for different *Aspergillus* and *Fusarium* toxins.

S/No	Mycotoxin	Products Ion (*m/z*)	Precursor Ion (*m/z*)	Q1 Pre Bias (V)	Collision Energy (CE)	Q3 Pre Bias (V)
1	OTA	239	403.8	−15	−27	−24
		221		−12	−38	−21
2	AFB_2_	259.10	315	−22	−31	−25
		287		−23	−26	−30
3	AFB_1_	241	313	−22	−41	−23
		285.1		−22	−24	−29
4	AFG_2_	245.1	331	−12	−32	−24
		313		−12	−24	−20
5	AFG_1_	243	329	−12	−28	−23
		313.1		−16	−24	−14
6	ZEN	185	319.1	−12	−27	−30
		187.1		−15	−21	−19
7	DON	231	297.10	−21	−13	−26
		249.10		−14	−12	−25

OTA: ochratoxin A; AFB_1_: aflatoxin B_1_; AFG_1_: aflatoxin G_1_; AFB_2_: aflatoxin B_2_; AFG_2_: aflatoxin G_2_; ZEN: zearalenone and DON: deoxynivalenol.

## Data Availability

The raw datasets obtained in this study are accessible upon request from the corresponding authors.

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
