# Peer review of "Seasonal and Geographical Impact on the Mycotoxigenicity of Aspergillus and Fusarium Species Isolated from Smallholder Dairy Cattle Feeds and Feedstuffs in Free State and Limpopo Provinces of South Africa"

_toxins, 2023, doi:10.3390/toxins15020128_

Round 1

Reviewer 1 Report

1. Although experiments were well performed, the data analysis somewhat lack depth. Can the collection of strains isolated from just two locations (small holder dairy cattle feeds and feedstuffs sampled) indicate that there are no significant geographical differences in toxin production?

2. The higher temperature conditions during summer seems to be more likely to fungal grow and mycotoxins production than in winter, so, the conclusion of high toxicity production in summer seems to be an established fact, where is the innovation of the article?

3. OTA production is affected by several environmental conditions such as temperature, Aw, and carbon conditions, the manuscript did not detect OTA in winter and summer, is the yields too low to detect or the culture condition is not suitable?

4. line 225-265 :Aspergillus spp can produce OTA in many cases, and it is recommended that references be cited to add depth to the discussion.

Insights into the underlying mechanism of ochratoxin A production in Aspergillus niger CBS 513.88 using different carbon sources.// Effect of temperature, water activity, and pH on growth and production of ochratoxin A by Aspergillus niger and Aspergillus carbonarius from Brazilian grapes.

5. the Ret. Time of AFB1 and AFB2 are 7.84 and 7.64, is the chromatogram well differentiated?

6. line 233 Gene names should be italicized. please check the full text.

7. line 280 FBs?

8. line 20010/11 season.?

9. line 327-329 “°C”! Note the writing of symbols! There should be no space between the number and °C, please check the full text. In addition,

10. Table 4 and 5  “a” Missing markers.

11. p-value. please check the full text.

12. the data lack the Statistical Analysis.

Author Response

The authors would like to thank the editor and reviewers for their constructive feedback on our manuscript, and to improve its quality. All the suggestions and recommendations have been considered and addressed by the authors. Where applicable, major additions to the manuscript were highlighted in yellow. Details of each correction/rebuttal is provided in the uploaded file.

Reviewer 2 Report

The manuscript investigated mycotoxin-producing potentials of a total of 112 Fusarium and Aspergillus strains obtained from smallholder dairy cow diets across different seasons from Free State and Limpopo provinces of SA, and also investigated the seasonal and geographical impact on the mycotoxin production capacity of the fungal strains. The results in this study lay the foundation for monitoring the mycotoxins in food and feed that can necessitates particular attention, especially for feed during the summer. However, some comments for the manuscript need to be addressed before it can be accepted.

1. In the section of “Introduction” (lines 67-69), the negative impacts of mycotoxin on animal were introduced. It is necessary to cite related studies, such as the following reports on the mycotoxin impact on the animals, to support the view herein:

Wang, Y.; Liu, F.; Zhou, X.; Liu, M.; Zang, H.; Liu, X.; Shan, A.; Feng, X. Alleviation of Oral Exposure to Aflatoxin B1-Induced Renal Dysfunction, Oxidative Stress, and Cell Apoptosis in Mice Kidney by Curcumin. Antioxidants 2022, 11, 1082. https://doi.org/10.3390/antiox11061082.

Ochieng, P.E.; Scippo, M.-L.; Kemboi, D.C.; Croubels, S.; Okoth, S.; Kang’ethe, E.K.; Doupovec, B.; Gathumbi, J.K.; Lindahl, J.F.; Antonissen, G. Mycotoxins in Poultry Feed and Feed Ingredients from Sub-Saharan Africa and Their Impact on the Production of Broiler and Layer Chickens: A Review. Toxins 2021, 13, 633. https://doi.org/10.3390/toxins13090633

2. In Table 1, The R2 value of OTA “0.99867” should be changed to 4 significant digits “0.9987”, and consistent with other R2 values in this column.

3. Line 123: “Supplementary Tables 1 and 2”, no Tables 1 and 2 in the supplementary files were provided. The number and proportion of strains in lines 123-129 cannot be checked. In this section, 2 significant digits of percentage value are not enough, more significant digits should be provided. For instance, in the sentence “of the 29 A. flavus strains examined in this work, 83 and 38% were positive for AFB1 and AFB2, respectively” in lines 130-131, the values of 83% and 38% are better givens as 82.76%, 37.93%.

4. In Tables 2 and 3, In row 2 of the tables, “(%)” is meaningless and confusing. In the columns of “No of Tox.spp”, it is necessary to add the percentage value besides of the number of strains.

5. In Tables 4 and 5, what’s the meaning of the superscript “a” on the word “Statistics”?

6. Line 215: “none produced”, it is a grammar mistake, please correct.

7. Line 221: “1048.71 μg/kg”, the highest concentration of AFB1 shown in Table 2 and 3 is 1045.80, which value is correct?

8. Line 292 and 294: what meaning of “2009/10 season” and “20010/11 season”? elaborate them please.

9. In line 453, The LODs were calculated as three times the ratio of the residual standard deviation to the slope, but in the Equation (line 467), the number “3.3” was used, rather than 3, which one is right?

10. Line 448” MgSO4·7H2O (5 g), and FeSO4·7H2O”, “H2O” should be “H2O”.

Author Response

(The authors gave the same response as above.)

Round 2

Reviewer 1 Report

I think we can accept in present form.

Reviewer 2 Report

The DOIs of some references including References 2, 4, 9, 11, 13, 17, 19, 20, 22, 33, etc. are missing,  add them please.